# Comparison of Fecal Microbiota Communities between Primiparous and Multiparous Cows during Non-Pregnancy and Pregnancy

**DOI:** 10.3390/ani13050869

**Published:** 2023-02-27

**Authors:** Xianbo Jia, Yang He, Zhe Kang, Shiyi Chen, Wenqiang Sun, Jie Wang, Songjia Lai

**Affiliations:** Farm Animal Genetic Resources Exploration and Innovation Key Laboratory of Sichuan Province, Sichuan Agricultural University, Chengdu 611130, China

**Keywords:** cow, pregnancy, primiparous and multiparous, fecal microbiota, 16S rRNA sequencing

## Abstract

**Simple Summary:**

An imbalance of the gut microbiota composition may lead to several reproductive disorders and physiological diseases during pregnancy. This study investigates the fecal microbiome composition between primiparous and multiparous cows during non-pregnancy and pregnancy to analyze the host-microbial balance at different stages. The results indicate that host-microbial interactions promote adaptation to pregnancy and will benefit the development of probiotics or fecal transplantation for treating dysbiosis and preventing disease development during pregnancy.

**Abstract:**

Imbalances in the gut microbiota composition may lead to several reproductive disorders and diseases during pregnancy. This study investigates the fecal microbiome composition between primiparous and multiparous cows during non-pregnancy and pregnancy to analyze the host-microbial balance at different stages. The fecal samples obtained from six cows before their first pregnancy (BG), six cows during their first pregnancy (FT), six open cows with more than three lactations (DCNP), and six pregnant cows with more than three lactations (DCP) were subjected to 16S rRNA sequencing, and a differential analysis of the fecal microbiota composition was performed. The three most abundant phyla in fecal microbiota were *Firmicutes* (48.68%), *Bacteroidetes* (34.45%), and *Euryarchaeota* (15.42%). There are 11 genera with more than 1.0% abundance at the genus level. Both alpha diversity and beta diversity showed significant differences among the four groups (*p* < 0.05). Further, primiparous women were associated with a profound alteration of the fecal microbiota. The most representative taxa included *Rikenellaceae_RC9_gut_group*, *Prevotellaceae_UCG_003*, *Christensenellaceae_R_7_group*, *Ruminococcaceae UCG-005*, *Ruminococcaceae UCG-013*, *Ruminococcaceae UCG-014*, *Methanobrevibacter*, *and [Eubacterium] coprostanoligenes group*, which were associated with energy metabolism and inflammation. The findings indicate that host-microbial interactions promote adaptation to pregnancy and will benefit the development of probiotics or fecal transplantation for treating dysbiosis and preventing disease development during pregnancy.

## 1. Introduction

Pregnancy is a wonderful and complex physiological process. In order to adapt to the growth and development of the fetus, drastic changes occur in maternal hormones, immunity, and metabolism before and after pregnancy. For mammals, progesterone (P4), estradioal (E2), follicle stimulating hormone (FSH), luteinizing hormone (LH), and Prolactin (PRL) are the main reproductive hormones to maintain and evaluate maternal pregnancy [1]. Growth hormone, thyroid hormone, and sex hormones could also change with maternal pregnancy [2]. The maternal immune system undergoes significant adaptations during pregnancy to avoid harmful immune responses against the fetus and to protect the mother and her future baby from pathogens [3]. For example, the number of T cells during pregnancy is lower than before pregnancy [4]. More nutrients are needed to be stored and consumed during pregnancy to meet the nutritional demands of the mother and fetus. Maternal metabolism changes to meet the nutritional requirements during pregnancy, the most obvious being the decrease in insulin sensitivity [5,6]. Additionally, compared to multiparous women, primiparous women have more exaggerated physiological responses, resulting in higher weight gain and body fat gain than that of multiparous women during pregnancy [7]. There are also many differences between primiparous and multiparous cows, including productivity, reproductive ability, energy balance, immune, metabolic, and hormonal responses [8,9].

Gut microbiota can produce a variety of nutrients, such as amino acids, fatty acids, and vitamins, which play an important role in regulating host metabolism, energy balance, and immune response [10,11,12,13]. With the changes of maternal hormones, immunity and metabolism during pregnancy, the composition and abundance of gut microbiota also shifted. The relative abundance of 21 genera of gut microbiota showed significant differences between non-pregnant and pregnant mice fed a standard diet. There were 4 abundant genera (present at greater than 1%) significantly increased and 5 rare taxa (present at lower than 0.5%) reduced during pregnancy compared to non-pregnant mice [14]. For dairy cows, the fecal microbial communities change dramatically in bacterial abundance at different taxonomic levels among the 12 distinctly defined production stages in a modern dairy farm, especially between virgin cows and parous cows [13]. Information on host-microbial interactions during pregnancy is emerging [15]. Recent studies showed that gut microbiota can impact the synthesis and metabolism of a variety of substances during pregnancy, regulating body weight, blood pressure, blood sugar, blood lipids, and other physiological indexes, and even leading to some pregnancy complications [16,17,18]. Parity has also been identified as one of the key determinants of the maternal microbiome during pregnancy. The difference in microbiome trajectories among different parities was significant in sows, with the greatest difference between zero parity and low parity animals. It was suggested that there are dramatic differences in the microbial trajectories of primiparous and multiparous animals [19]. Compared to multiparous sows, primiparous sows had a lower gut microbiota richness and evenness during the periparturient period [20]. Primiparous cows have different uterine and rumen microbiome compositions compared to multiparous cows [21,22]. However, it is still unclear if parity impacts the maternal cow’s gut microbiome during both non-pregnancy and pregnancy.

In this study, the gut microbiome composition was investigated in fecal samples from primiparous and multiparous cows during non-pregnancy and pregnancy. It confirmed that there is an inherent shift in gut microbiota associated with pregnancy and differences in gut microbiota composition between primiparous and multiparous animals. The results will help develop strategies to improve the reproductive management of cows.

## 2. Materials and Methods

### 2.1. Ethics Statement

The collection of biological samples and experimental procedures carried out in this study were approved by the Institutional Animal Care and Use Committee in the College of Animal Science and Technology, Sichuan Agricultural University, China (DKY20210306).

### 2.2. Sample Collection

A total of 24 healthy Holstein cows were selected from one dairy herd under the same conditions in southwestern China, with the same feeding processes, similar body conditions, and similar body weight. According to their reproductive stages, the cows were divided into four groups: the cows before their first pregnancy (13 months, *n* = 6, BG); at their first pregnancy (the 4th month of pregnancy, 18 months, *n* = 6, FT); open cows with more than three lactations (30 days after parturition, 57 months, *n* = 6, DCNP); and pregnancy cows with more than three lactations (the 4th month of pregnancy, 60 months, *n* = 6, DCP). Animals were fed the total mixed ration (TMR) made according to NRC (2012) with the same feed raw material. None of the cows had received antibiotics in the last 3 months. All 24 fecal samples were obtained once from cow rectum content on the same day, transferred to separate sterilized 2 mL tubes, and stored immediately in liquid nitrogen. All samples were then transported to the laboratory and stored at −80 °C for further analysis.

### 2.3. DNA Extraction, PCR Amplification and Gene Sequencing

Total genome DNA was extracted from fecal samples, the negative control (DNA free water), and the positive control (16S Universal E29), using a BIOMICS DNA Microprep Kit (Zymo Research, D4301, Irvine, CA, USA) according to the manufacturer’s instructions. DNA concentration and purity were tested on 0.8% agarose gels. DNA yield was detected with a Tecan Infinite 200 PRO fluorescent reader (Tecan Systems Inc., San Jose, CA, USA). The 16S rRNA amplification covering the variable region V4-V5 was carried out using the primers 338F (5′-ACTCCTACGGGAGGCAGCAG-3′) and 915R (5′-GTGCTCCCCCGCCAATTCCT-3′) by a Thermal Cycler PCR system (Gene Amp 9700, ABI, Foster City, CA, USA). PCRs were performed in triplicate in a 25 µL mixture. The PCR products were diluted six times, quantified with electrophoresis on 2% agarose gel, and then purified by the Zymoclean Gel Recovery Kit (Zymo Research, D4008, Irvine, CA, USA). About 100 ng of DNA were used for library preparation. The library was prepared using the TruSeq^®^ DNA PCR-Free Sample Preparation Kit (Illumina, San Diego, CA, USA), followed by quality evaluation on the Qubit@ 2.0 Fluorometer (Thermo Fisher Scientific, Waltham, MA, USA) and Agilent Bioanalyzer 2100 system (Agilent, Santa Clara, CA, USA). Library was finally paired-end sequenced (2 × 300) on an Illumina MiSeq PE300 platform (Illumina, San Diego, CA, USA).

### 2.4. Data Analysis

The raw fastq files were merged using FLASH [23]. The raw tags were analyzed using the QIIME (v1.9.0) pipeline [24]. All tags were quality filtered. Sequences shorter than 200 nt with an average quality value less than 25, and those containing two or more ambiguous bases, were discarded. The clean tags were then mapped to the Gold database (http://drive5.com/uchime/uchime_download.html (accessed on 5 May 2021)) using UCHIME algorithm, followed by removal of the chimera sequences to identify the effective tags [25]. The operational taxonomic units (OTUs) table was created at 97% similarity using the UPARSE pipeline [26]. Representative sequences from each OTU were aligned to 16S reference sequences with PyNAST [27]. The phylogenetic trees were drawn using FastTree [28]. Annotation analysis was performed using the UCLUST taxonomy and the SILVA database [29,30].

The abundance of OTUs was normalized using a standard sequence number corresponding to the sample with the least sequence. The comparison of OTU numbers used a one-way analysis of variance (one-way ANOVA), followed by the Bonferroni multiple comparisons test. The alpha diversity was calculated to analyze the complexity of species diversity in the sample, including observed species, Chao1, Shannon, Simpson, coverage, and Faith’s PD. The beta diversity, weighted Unifrac and unweighted UniFrac, was calculated to evaluate the differences of samples in species complexity. Principal coordinate analysis (PCoA) was used to visualize differences in bacterial community composition among groups. The linear discriminant analysis coupled with effect size (LEfSe) was performed to identify the differentially abundant taxa between different groups. Pairwise comparisons were made using metagenomSeq.

## 3. Results

### 3.1. Sequencing Information

In order to evaluate the effect of reproductive status on the cow fecal microbiota, the V4–V5 hypervariable regions of the 16S rRNA gene were sequenced in the microbial communities of 24 samples. A total of 705,988 raw PE reads were generated from these 24 samples (average: 29,416 ± 4914, range: 21,956–36,765). After quality control, 632,192 effective tags were obtained from 24 samples (average: 26,341 ± 4408, range: 19,472–32,926), with an average of 407.67 ± 0.92 bps per tag after the merging of overlapping paired-reads, quality filtering, and removing of chimeric sequences. By the 97% sequence similarity, 6842 OTUs were computationally constructed with 1727.38 ± 405.39 (range: 999–2788) as the mean number of OTUs per sample, and the mean number of OTU in DCNP group was significantly lower than that of BG and FT group (*p* < 0.01) (Figure 1).

### 3.2. Microbial Ecology of the Fecal Microbiome

These 6842 OTUs taxonomically assigned to microbial 2 Kingdom, 17 phyla, 25 classes, 38 orders, 67 families, 168 genera, and 117 species. According to OTUs’ number, the average abundance of each group at different category levels was evaluated (Figure 2). The fecal microbial communities were dominated by bacteria (84.58%), and archaea were only 15.42% abundant. The most abundant phyla across all 24 metagenomic libraries were *Firmicutes* (48.68%), followed by *Bacteroidetes* (34.45%), and *Euryarchaeota* (15.42%). Other less abundant phyla were *Spirochaetes* (0.85%), *Tenericutes* (0.42%), *Proteobacteria* (0.07%), *Actinobacteria* (0.06%), *Fibrobacteres* (0.02%), *Cyanobacteria* (0.02%), and *Planctomycetes* (0.01%) (Figure 3). At the genus level, there are 11 genera with more than 1.0% abundance, including *Ruminococcaceae UCG-005* (21.91%), *Methanobrevibacter* (13.28%), *Rikenellaceae RC9 gut group* (10.13%), *[Eubacterium] coprostanoligenes group* (7.10%), *Prevotellaceae UCG-004* (6.47%), *Alistipes* (5.52%), *Ruminococcaceae UCG-013* (4.89%), *Prevotellaceae UCG-003* (4.61%), *Ruminococcaceae UCG-014* (1.78%), *Methanocorpusculum* (1.42%), *Christensenellaceae R-7 group* (1.12%) (Figure 4).

### 3.3. Microbial Diversity of the Fecal Microbiome

The alpha diversity indexes, including observed species, Chao1, Shannon, Simpson, coverage, and Faith’s PD, for four groups were calculated to estimate species richness and diversity (Figure 5). Compared to the BG and FT groups, the observable species, Chao1, and Faith’s PD were significantly lower, and coverage was significantly higher in the DCNP group (*p* < 0.05, Kruskal–Wallis test), but without statistical significance in the DCP group (*p* > 0.05, Kruskal–Wallis test). Further, no statistically significant difference was shown among the four groups in Shannon and Simpson (*p* > 0.05, Kruskal–Wallis test).

Based on the Jaccard and Bray–Curtis methods, principal coordinated analysis (PCoA) of beta diversity was further used to analyze compositional differences in fecal microbiota among four groups (Figure 6). The samples in the BG, FT, and DCP groups were clustered together according to their particular groups, while the samples in the DCNP group were spread out. The samples in the BG and FT groups tended to cluster together in accordance with PCoA results. Both Jaccard and Bray-Curtis distances showed significant differences among the four groups (ANOSIM, *p* < 0.01), except between groups DCP vs. DCNP (ANOSIM, *p* > 0.05).

### 3.4. Microbial Taxonomy and Function Analysis

Linear discriminant analysis effect size (LEfSe) was used to discover the differential microbiota and estimate their effect size. Based on LEfSe, it restrictively analyzed the successfully annotated species and detected 60 taxa significantly different in abundance among four groups. There were 7 taxa significantly more abundant in the BG group, 17 in the FT group, 8 in the DCNP group, and 28 in the DCP group (Figure 7). The most representative taxa were *Rikenellaceae* and *Rikenellaceae_RC9_gut_group* in the DCP group, *Prevotellaceae* and *Prevotellaceae_UCG_003* in the FT group, *Christensenellaceae_R_7_group* in the DCNP group, and *Firmicutes, Clostridia, Clostridiales, and Ruminococcaceae* in the BG group.

The metagenomeSeq was further used to compare the abundance of OTUs between each group. The abundance of 4, 12, and 23 OTUs was significantly increased, while that of 1, 2, and 17 OTUs was significantly reduced in the FT, DCNP, and DCP groups compared with the BG group, respectively. In the three comparison groups, the abundance of six common genera (>1%), namely *Prevotellaceae UCG-003, Ruminococcaceae UCG-013, [Eubacterium] coprostanoligenes group, Rikenellaceae RC9 gut group, Methanobrevibacter*, and *Ruminococcaceae UCG-005*, was identified as a significant difference (Figure 8). There were 16 and 21 OTUs that were significantly increased, and 2 and 19 OTUs that were significantly reduced, in the DCNP and DCP groups compared with the FT group, respectively. A total of 8 common genera, such as the *Christensenellaceae R-7 group, Ruminococcaceae UCG-014, Prevotellaceae UCG-003, Ruminococcaceae UCG-013, [Eubacterium] coprostanoligenes group, Rikenellaceae RC9 gut group, Methanobrevibacter*, and *Ruminococcaceae UCG-005*, were observed to have significant differences (Figure 8). Furthermore, in the DCP group, the abundance of 4 OTUs decreased compared with the DCNP group. The relative abundance of 2 common genera, *Methanobrevibacter and Prevotellaceae UCG-003*, in the DCNP group was higher than that in the DCP group.

## 4. Discussion

The reproductive efficiency and health of cows have always been priorities. The gut microbiota composition plays an important role in the reproductive performance throughout a female’s lifetime. In humans, the gut microbiome has been considered to affect every stage and level of female reproduction, including follicle and oocyte maturation in the ovary, fertilization and embryo migration, implantation, the whole pregnancy, and parturition [31,32,33,34]. The gut microbial communities can influence reproductive success from mate choice to healthy pregnancy and successfully producing offspring in animals [35,36]. Recent studies reported that bovine vaginal and fecal microbiome associated with differential pregnancy outcomes [37,38]. The fecal microbiome predicted pregnancy with a higher accuracy than that of the vaginal microbiome [38]. In this study, the fecal microbiota were investigated in 4 different reproductive stages and revealed the dramatic changes in fecal microbiota diversity and composition among 4 groups using the sequencing of the 16S rRNA gene. 

In this study, *Firmicutes, Bacteroidetes*, and *Euryarchaeota* were the three most dominant phyla, and *Ruminococcaceae UCG-005, Methanobrevibacter, and Rikenellaceae RC9 gut group* were the three most dominant genera in the cow fecal samples. They were consistent with several earlier studies [39]. In previous studies, *Bacteroidetes* (51.6~59.74%) and *Firmicutes* (27.6~38.74%) together comprised up to 81.6~93.20% of the cow fecal bacterial abundance [13,40,41]. The phylum *Euryarchaeota* was predominant within the Archaea and accounted for around 0.25% of the cow fecal microbiota abundance [41,42]. *Ruminococcaceae UCG-005, Methanobrevibacter*, and *Rikenellaceae RC9 gut groups* predominate in the *Firmicutes, Euryarchaeota*, and *Bacteroidetes* phyla, respectively. *Ruminococcaceae UCG-005* and *Rikenellaceae RC9 gut group* usually had a relative abundance >8% of fecal microbiota in dairy cows. The genus *Methanobrevibacter* comprised more than 80% of the phylum *Euryarchaeota* in cow fecal Archaea [13,43].

The age and pregnancy are two important factors contributing to the species richness and diversity of fecal microbiota. The alpha diversity index, observed species, Chao 1, coverage, and Faith’s PD were significantly different among the BG, FT, and DCNP groups in this study. However, the cluster among four groups was significant, separating BG and FT groups from DCNP and DCP groups by PCoA based on Jaccard and Bray–Curtis distances. These also showed that the greatest differences in microbiome trajectories occurred between nulliparous and primiparous animals [19]. Nulliparous animals had higher gut microbial diversity than that of primiparous animals, and pregnancy could increase gut microbial diversity [19,20]. The effect of age is more related to calving. The increase in alpha diversity during pregnancy could be due to an increase in nutrient requirements during lactation. The first birth is the most important physiological change in a cow’s life, and pregnancy increases metabolism.

In order to further identify important taxa differed among groups, LEfse and metagenomeSeq analyses were conducted. LEfse analysis is helpful to discover the important differential taxa (biomarkers) and estimate their effect sizes. The LEfSe analysis revealed that the most differentially abundant taxa were in DCP, followed by FT, DCNP, and BG. The metagenomeSeq analyses showed that the comparisons with the most significant differences in microbial taxa are BG vs. DCP and FT vs. DCP, followed by FT vs. DCNP, BG vs. DCNP, BG vs. FT, and DCNP vs. DCP. These suggested that parturition experience is one of the most important factors to impact cattle gut microbiome trajectory. Previous study also reported that the most difference in microbiome trajectory occurred between nulliparous and low parity sows [19]. There was significant difference between multiparous and primiparous cows on vaginal and uterine microbiotas [44,45]. The most representative taxa were associated with energy metabolism and inflammation. Mice fed with high-fat diet increased the richness of gut microbial *Rikenellaceae_RC9_gut_group*. The high-fat diet also increased the risks of intestinal pathogen colonization and inflammation [46]. Supplementation of probiotics increased the relative abundance of *Prevotellaceae_UCG_003*, which improved the energy status of the beef steers [47]. Fibrolytic enzyme increased the relative abundance of *Christensenellaceae_R_7_group*, which improve the average daily gain and feed conversion ratio of lambs [48]. The ruminococcaceae family is the predominant acetogen in the cattle rumen, which is related to cellulose and hemicellulose degradation [49]. The carbohydrate resource and the fiber decomposition process in diet contribute to the different abundances of *Ruminococcaceae UCG-005, Ruminococcaceae UCG-013, Ruminococcaceae UCG-014*, and other *Ruminococcaceae* in cattle feces [49,50]. Methanobrevibacter is another common inhabitant of the cattle rumen, which can reduce CO_2_ with H_2_ to form methane [51,52].The serum cholesterol concentration tended to be lower after feeding *Eubacterium* coprostanoligenes to germ-free mice [53]. Thus, gut microbes are involved in changes in energy intake and immunity during cattle adaption to pregnancy.

## 5. Conclusions

In conclusion, this study investigated the difference in fecal bacterial communities between primiparous and multiparous cows during non-pregnancy and pregnancy. The results revealed that pregnancy increased the relative abundance and diversity of fecal microbiota, while aging reduced those traits. In addition, primiparous were related to a profound alteration of the fecal microbiota. The most representative taxa included *Rikenellaceae_RC9_gut_group, Prevotellaceae_UCG_003, Christensenellaceae_R_7_group, Ruminococcaceae UCG-005, Ruminococcaceae UCG-013, Ruminococcaceae UCG-014, Methanobrevibacter, and [Eubacterium] coprostanoligenes group*, which were associated with energy metabolism and inflammation. In the future, further functional studies will be able to treat dysbiosis and prevent disease development during pregnancy by using probiotics or fecal transplantation.

## Figures and Tables

**Figure 1 animals-13-00869-f001:**
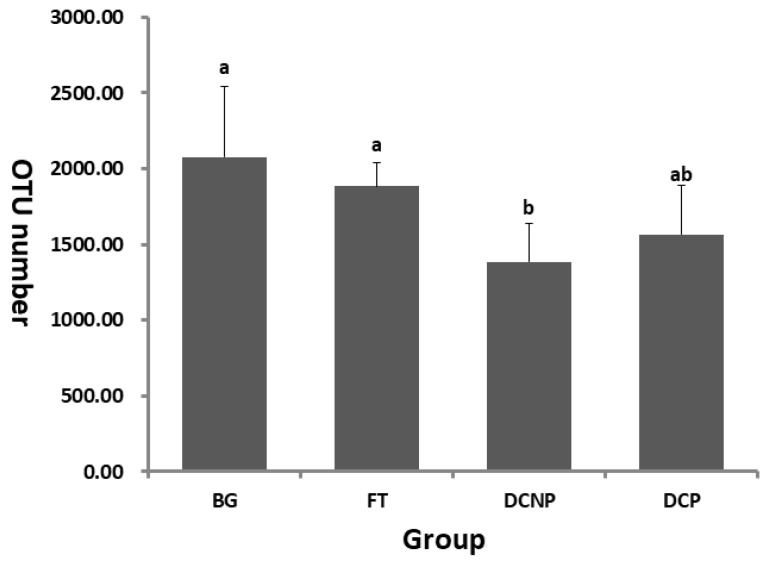
Histogram of the number of OTUs in the four groups. Different letters indicate a significant difference among groups (*p* < 0.05), while the same letter indicates no difference among groups (*p* > 0.05).

**Figure 2 animals-13-00869-f002:**
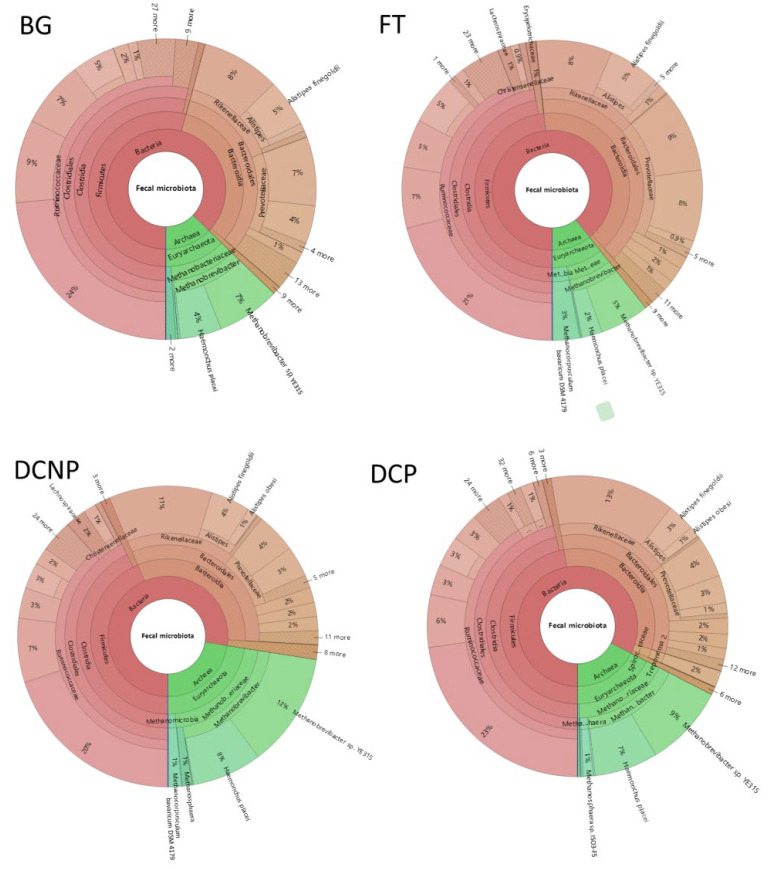
The fecal microbiota composition of the four groups.

**Figure 3 animals-13-00869-f003:**
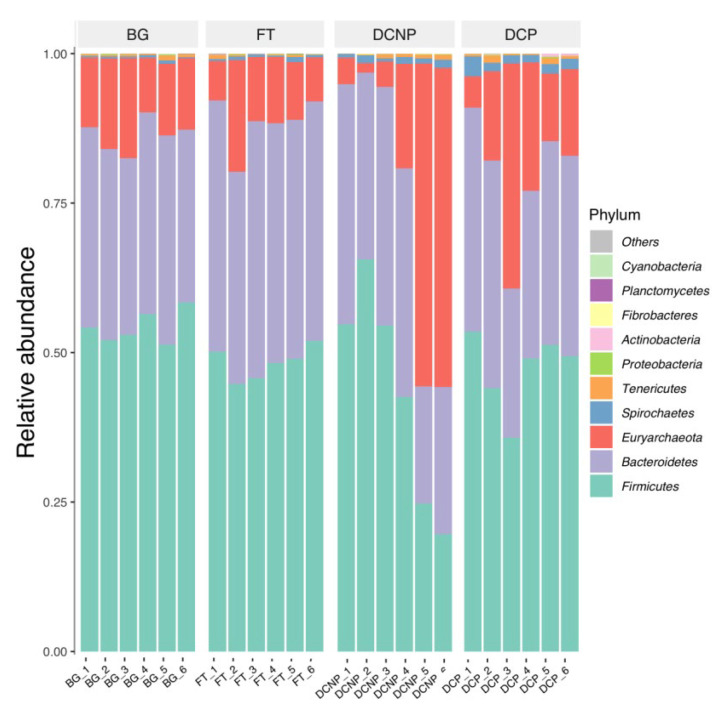
Relative abundance of fecal microbiota at the phylum level in the four groups.

**Figure 4 animals-13-00869-f004:**
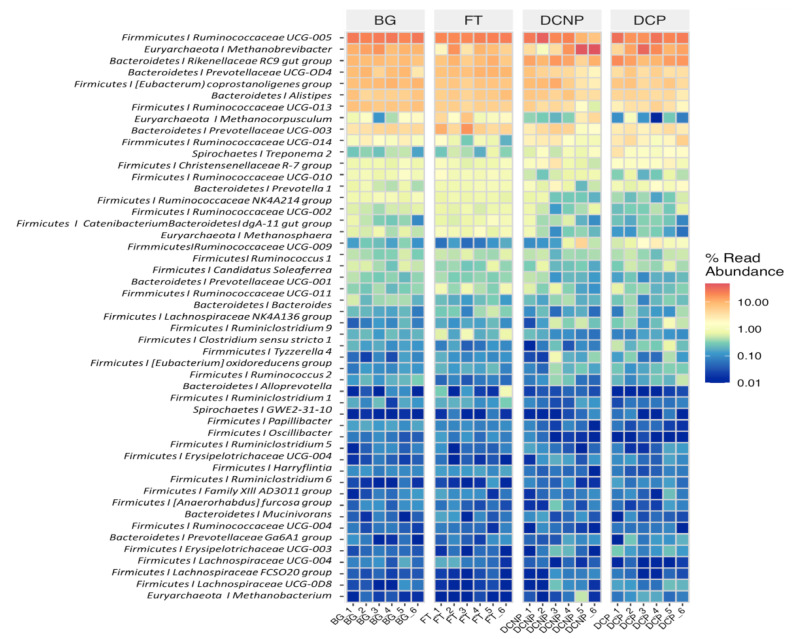
Heatmap of high abundance at genus level in the four groups.

**Figure 5 animals-13-00869-f005:**
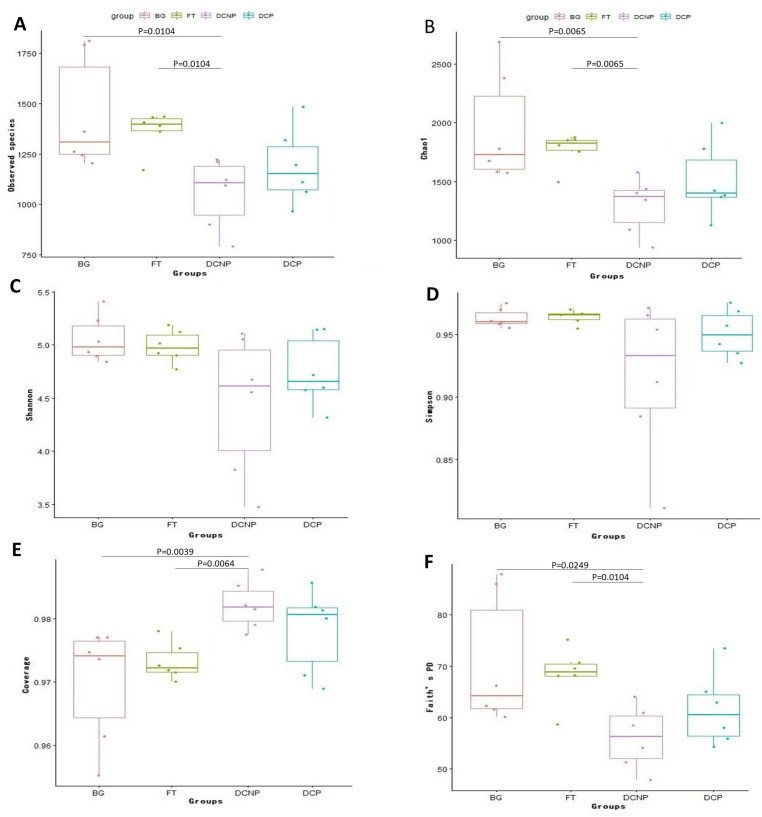
Box-plot representation of alpha diversity among four groups. The number of observed OTUs (**A**), Chao1 (**B**), Shannon (**C**), Simpson (**D**), Coverage (**E**) and Faith’s PD (**F**) among the four groups.

**Figure 6 animals-13-00869-f006:**
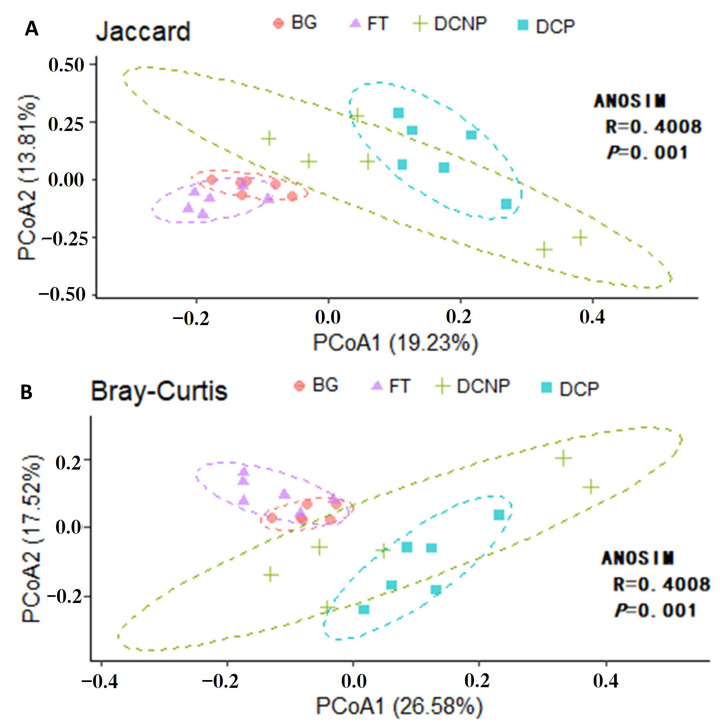
Principal Coordinates Analysis (PCoA) using Jaccard distance (**A**) and Bray–Curtis distance (**B**) among the four groups.

**Figure 7 animals-13-00869-f007:**
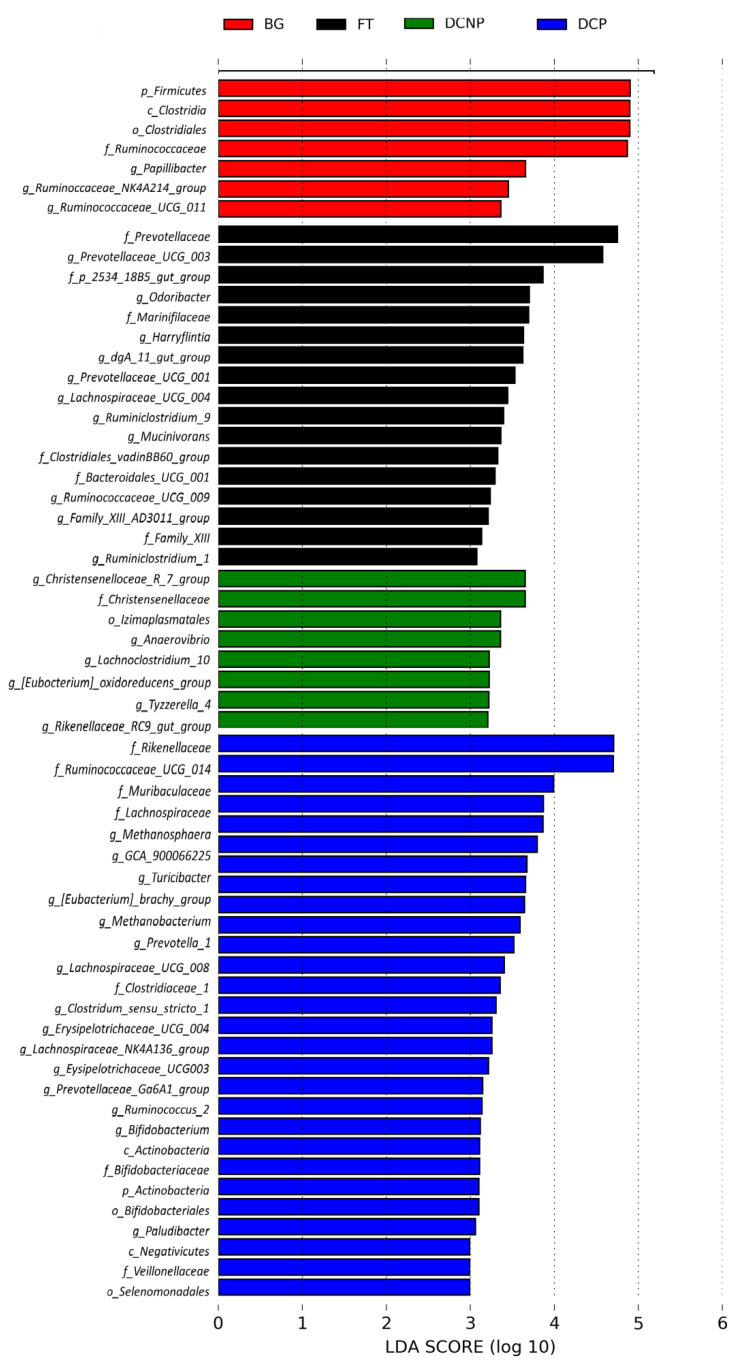
Histogram of the LDA scores for differentially abundant of fecal bacteria among the four groups.

**Figure 8 animals-13-00869-f008:**
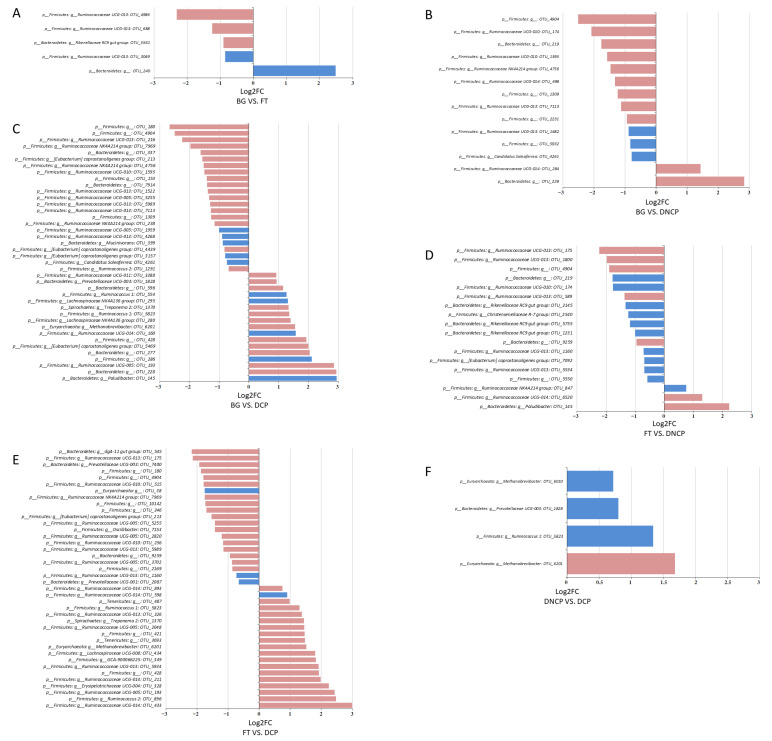
The differentially enriched OTUs among the four groups. Adjusted *p* < 0.05 (**blue**) or *p* < 0.01 (**red**) were considered significant. (**A**)-BG VS. FT, (**B**)-BG VS. DNCP, (**C**)-BG VS. DCP, (**D**)-FT VS. DNCP, (**E**)-FT VS. DCP, (**F**)-DNCP VS. DCP.

## Data Availability

The data presented in this study are available on request from the corresponding author.

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
