# Peer review of "Comparison of Fecal Microbiota Communities between Primiparous and Multiparous Cows during Non-Pregnancy and Pregnancy"

_animals, 2023, doi:10.3390/ani13050869_

Round 1
Reviewer 1 Report
It’s a nice study by authors investigating the fecal microbiome composition between primiparous and multiparous cows during non-pregnancy and pregnancy that helps treat dysbiosis and prevent disease development during pregnancy. However, I will suggest some minor points for improvement before the final publication.
-Formatting required.
-Double-check the references in the texts and citation section and vice versa.
Line 17, remove the word “physiological”
In the material and method sections, please explain 24 healthy Holstein cows used in this study. The sampling was done at the same time from time to time on their availability as a grouping.
Lie 98 Zymo Research country?
-The font size and style in the figure are not the same.
-All the figures, please enhance the quality of the figure. Especially readable.
-please italicize all scientific words where it's applicable.
-Please cite the below paper that might be relevant to your study.
Effects of Probiotics and Gut Microbiota on Bone Metabolism in Chickens: A Review.
Integrated Fecal Microbiome and Metabolomics Reveals a Novel Potential Biomarker for Predicting Tibial Dyschondroplasiain Chickens.
Author Response
Comments and Suggestions for Authors
It’s a nice study by authors investigating the fecal microbiome composition between primiparous and multiparous cows during non-pregnancy and pregnancy that helps treat dysbiosis and prevent disease development during pregnancy. However, I will suggest some minor points for improvement before the final publication.
-Formatting required.
Re:We checked them.
-Double-check the references in the texts and citation section and vice versa.
Re:We checked them.
Line 17, remove the word “physiological”
Re:We removed the word "physiological".
In the material and method sections, please explain 24 healthy Holstein cows used in this study. The sampling was done at the same time from time to time on their availability as a grouping.
Re:All 24 fecal samples were obtained once from cow rectum content at the same day, and transferred to separate sterilized 2 mL tubes, and stored immediately in liquid nitrogen.
Lie 98 Zymo Research country?
Re:ZYMO RESEARCH is a biological company located in Orange County, California, USA.
-The font size and style in the figure are not the same.
Re:We modified the font size and style of the pictures.
-All the figures, please enhance the quality of the figure. Especially readable.
Re:We enhance the quality of the figures.
-please italicize all scientific words where it's applicable.
Re:We modified them.
-Please cite the below paper that might be relevant to your study.
Effects of Probiotics and Gut Microbiota on Bone Metabolism in Chickens: A Review.
Integrated Fecal Microbiome and Metabolomics Reveals a Novel Potential Biomarker for Predicting Tibial Dyschondroplasiain Chickens.
Re:We cited these two papers.
Reviewer 2 Report
Estimate editor,
I have reviewed the article “Comparison of fecal microbiota communities between primiparous and multiparous cows during nonpregnancy and pregnancy “ . After trying to make sense of the research, I feel compelled to reject it.
This study investigates the fecal microbiome composition between primiparous and multiparous cows during non-pregnancy and pregnancy to analyze the host-microbial balance at different stages.
They have detected that there are differences in the microbiota in pregnant cows compared to non-pregnant cows. And this conclusion is very poor, because the metabolic and hormonal status of a pregnant cow is different from the status of a non-pregnant cow. Therefore, it is normal for the microbiota to be different.
They should try to determine what consequences this change has at the level of intestinal microbiota on reproductive efficiency (in ovarian folliculogenesis, or at the level of embryonic development, etc...) Because such a change detected in the microbiota, it is possible that it is normal and necessary due to the pregnant status of the animal.
The title of the manuscript is very attractive, but when one studies the article, the title of the article does not correspond to the development of the article.
For all these reasons, I am obliged not to accept the manuscript, as it is presented. If the authors continue to develop this line of research, the implication of the microbiota in pregnant women on reproductive efficiency, a very interesting and publishable article will surely emerge.
Author Response
Comments and Suggestions for Authors
Estimate editor,
I have reviewed the article “Comparison of fecal microbiota communities between primiparous and multiparous cows during nonpregnancy and pregnancy “ . After trying to make sense of the research, I feel compelled to reject it.
This study investigates the fecal microbiome composition between primiparous and multiparous cows during non-pregnancy and pregnancy to analyze the host-microbial balance at different stages.
They have detected that there are differences in the microbiota in pregnant cows compared to non-pregnant cows. And this conclusion is very poor, because the metabolic and hormonal status of a pregnant cow is different from the status of a non-pregnant cow. Therefore, it is normal for the microbiota to be different.
They should try to determine what consequences this change has at the level of intestinal microbiota on reproductive efficiency (in ovarian folliculogenesis, or at the level of embryonic development, etc...) Because such a change detected in the microbiota, it is possible that it is normal and necessary due to the pregnant status of the animal.
The title of the manuscript is very attractive, but when one studies the article, the title of the article does not correspond to the development of the article.
For all these reasons, I am obliged not to accept the manuscript, as it is presented. If the authors continue to develop this line of research, the implication of the microbiota in pregnant women on reproductive efficiency, a very interesting and publishable article will surely emerge.
Re:This study investigates the fecal microbiome composition between primiparous and multiparous cows during non-pregnancy and pregnancy to analyze the host-microbial balance at different stages.The results indicate that host-microbial interactions promote the adaptation to pregnancy.Of course,these findings are preliminary results.Further in-depth analysis is required on the basis of this study. And then, the results will benefit the development of probiotics or fecal transplantation for treating dysbiosis and preventing disease development during pregnancy.
Reviewer 3 Report
Summary
The aim of the study is to investigate the gut microbiota of pregnant and non-pregnant multiparous and primiparous cows. Faecal samples of six Holstein cows per group were analysed using the 16S rRNA sequencing of the V4-V5 region. In general, it was found that alpha diversity was higher in nulliparous (FG) and primiparous (FT) compared to multiparous non-pregnant cows (DCNP), but not significantly different compared to multiparous pregnant cows (DCP). Beta diversity analysis also showed different clustering patterns, where FG and FT samples clustered together, whereas DCNP were spread out. Several bacterial groups were found to be significantly different between groups. All these results suggest that age and pregnancy status may influence the gut microbiota composition in cows.
General comments
Thank you for conducting this study. Your study gives information about the relationship between reproductive status and gut microbiota composition that could be used in the future for the development of potential interventions based on gut microbiota. However, a more in-depth analysis is required.
Pregnancy, partum, and lactation are associated with profound changes not only at the hormonal level but also at the metabolic and immunological levels. Samples in pregnant cows were collected at mid-term, why did you choose this time point? While some studies have reported no significant changes during pregnancy in people, other studies have shown differences in the gut microbiota composition between early and late pregnancy. The transition from the dry period to high-performance milk production also imposes a huge challenge for cows. It is known that the body condition score, body weight, diet, milk yield, etc., influence reproductive performance and probably the gut microbiota composition. None of these factors was explained. Farm practices are not clear in this study. For open cows, you chose the time point 30 days post-partum, why did you choose this time point?
You found that 7 taxa were significantly more abundant in the BG group, 17 in the FT group, 8 in the DCNP group and 28 in the DCP group. However, in the discussion, you focused your analysis on only a few taxa, and it is not clear how the findings reported in other studies correlate with this study and these experimental conditions. For example, you have mentioned that Rikenellaceae_RC9_gut_group is associated with the consumption of a high-fat diet in mice, but this finding is not correlated with the present study (unless the cows were fed a high-fat diet). In fact, this taxon was only significantly increased in the DCP group. What would be the relevance of this finding in the present study?
Many dairy cows experience a state of energy deficit as they transition from late gestation to early lactation. These changes are characterised by increased lipolysis, and loss of sensitivity to insulin, among other changes. Could those changes be correlated with the bacterial composition during this period?
There is no need to describe each taxon, but the significance and relevance of these changes should be clearly explained in the discussion.
You also mention that the number of pregnancies influences the gut microbial composition, but you omit reasons for this finding. Could you please explain the reason for this?
Also, microbiome studies should include positive and negative controls during the DNA extraction and the library preparation. The low number of animals per group is also a limitation. Why did you choose 6 cows per group? Due to the inter-individual variation (that has been reported not only in people but also in cows), ideally, longitudinal studies are preferred. Following each cow over time during different reproductive stages could give a better overview of the changes in the gut microbiota over time. Limitations and recommendations for future studies are not included as they should be.
Metadata and sequencing results should be made available in a public repository.
Specific comments
Line 20: Why did you choose those acronyms to represent each group? What do the letters stand for?
Line 56 -59: Please review the sentence, changes in the writing are required for clarity. Also, it would be advisable to include the results of previous studies made with cows.
Line 86: Please provide more information about the study population, living conditions, farm practices, vaccination, environment, and diet. The age of each group should be specified as well as the body condition score and body weight. All these factors can influence gut microbiota composition. Also, for reproducibility, all these factors should be mentioned.
Line 97: How much faecal material was used for DNA extraction? Were they aliquoted before storage at -80, or were they thawed before DNA extraction? How much DNA was used for library preparation?
Line 112: Do you mean that the technical replicates were merged?
Line 123: did you perform rarefaction curves to assess if sequencing depth was appropriate?
Line 144: Spelling mistake in legend (umber versus number). Could you explain the statistics performed in figure 1, please? It was not specified in the methodology. Exact p-values should be specified.
Line 148: That number of genera and species probably doesn’t reflect the total number, as with 16S rRNA sequencing, not all genera and species can be identified.
Line 162: Two cladograms are shown for the FT group or there is a spelling mistake, and the third cladogram corresponds to DCNP.
Line 163: Visually inspecting figure 3, it is evident that DCNP_5 and _6 have a different profile compared to the other four cows, probably influencing the results, as only 6 cows were analysed by group. At least, at the phylum level. This is also probably reflected in figure 6, where it is evident that 4 cows cluster together whereas the other 2 clustered but separated from the first 4.
Line 247: Pregnancy status. If Age and pregnancy are contributing factors, why do you think there were no differences between multiparous pregnant cows and nulliparous and primiparous cows, as they differ in age and pregnancy status? Scientific conclusions should not be based only on whether a p-value passes a specific threshold. The number of subjects per group is small. Based on figure 2, in general, alpha diversity was higher in those of a younger age. Could you explain how age influences diversity, please?
Line 254: You mention that pregnancy is related to an increase in alpha diversity, could you explain the reason, please?
Line 267: You have used two different statistical methods for differential abundance. LEfSe was used for general comparison whereas pairwise comparisons were made using metagenomSeq (Based on figure 8). What was the reason? Currently, some people recommend the use of different statistical methods to find the most robust and consistent changes, but in this case, it is not clear why you decided to use two different statistical methods.
Author Response
The aim of the study is to investigate the gut microbiota of pregnant and non-pregnant multiparous and primiparous cows. Faecal samples of six Holstein cows per group were analysed using the 16S rRNA sequencing of the V4-V5 region. In general, it was found that alpha diversity was higher in nulliparous (FG) and primiparous (FT) compared to multiparous non-pregnant cows (DCNP), but not significantly different compared to multiparous pregnant cows (DCP). Beta diversity analysis also showed different clustering patterns, where FG and FT samples clustered together, whereas DCNP were spread out. Several bacterial groups were found to be significantly different between groups. All these results suggest that age and pregnancy status may influence the gut microbiota composition in cows.
General comments
Thank you for conducting this study. Your study gives information about the relationship between reproductive status and gut microbiota composition that could be used in the future for the development of potential interventions based on gut microbiota. However, a more in-depth analysis is required.
Pregnancy, partum, and lactation are associated with profound changes not only at the hormonal level but also at the metabolic and immunological levels. Samples in pregnant cows were collected at mid-term, why did you choose this time point? While some studies have reported no significant changes during pregnancy in people, other studies have shown differences in the gut microbiota composition between early and late pregnancy.The transition from the dry period to high-performance milk production also imposes a huge challenge for cows. It is known that the body condition score, body weight, diet, milk yield, etc., influence reproductive performance and probably the gut microbiota composition. None of these factors was explained. Farm practices are not clear in this study. For open cows, you chose the time point 30 days post-partum, why did you choose this time point?
Re: According to recent study, the gut microbiota of dairy cows is relatively stable during the 60~250 days of gestation, which is the most representative of the gut microbiota of pregnant cows.And the gut microbiota of open cows was relatively stable from 21 to 60 days postpartum, which was the best representative of the gut microbiota of open cows. So, we choosed these two time point. Many factors influence influence reproductive performance and probably the gut microbiota composition. When selecting samples, we consulted their production records and tried to select relatively consistent cows to carry out the experiment.
You found that 7 taxa were significantly more abundant in the BG group, 17 in the FT group, 8 in the DCNP group and 28 in the DCP group. However, in the discussion, you focused your analysis on only a few taxa, and it is not clear how the findings reported in other studies correlate with this study and these experimental conditions. For example, you have mentioned that Rikenellaceae_RC9_gut_group is associated with the consumption of a high-fat diet in mice, but this finding is not correlated with the present study (unless the cows were fed a high-fat diet). In fact, this taxon was only significantly increased in the DCP group. What would be the relevance of this finding in the present study?
Re:We focus on the possible function of these microbial taxa, trying to explain the reason of significant enrichment in different groups.For example, Rikenellaceae_RC9_gut_group assocaite with intestinal pathogens colonization and inflammation in mice gut. Therefore, we will not focus on the bacteria taxa whose function has not been reported.
Many dairy cows experience a state of energy deficit as they transition from late gestation to early lactation. These changes are characterised by increased lipolysis, and loss of sensitivity to insulin, among other changes. Could those changes be correlated with the bacterial composition during this period?
Re:In our study, both Jaccard and Bray-Curtis distances showed no significant differences between group DCP and group DCNP (ANOSIM, p>0.05).This is consistent with recent report. So, these changes may not be correlated with the bacterial composition during this period.
There is no need to describe each taxon, but the significance and relevance of these changes should be clearly explained in the discussion.
Re:Your suggestion is very good and we have revised it.
You also mention that the number of pregnancies influences the gut microbial composition, but you omit reasons for this finding. Could you please explain the reason for this?
Re: The age and pregnant are two important factors contributing to the species richness and diversity of fecal microbiota. The alpha diversity indexes, Observed species, Chao1, Coverage and Faith's PD, were significantly different among BG, FT and DCNP groups in this study. And the cluster among four groups was significant separate BG and FT groups from DCNP and DCP groups by PCoA based on Jaccard and Bray-Curtis distances. Nulliparous animals had higher gut microbial diversity than that of primiparous animals, and pregnancy could increase gut microbial diversity.
Also, microbiome studies should include positive and negative controls during the DNA extraction and the library preparation. The low number of animals per group is also a limitation. Why did you choose 6 cows per group? Due to the inter-individual variation (that has been reported not only in people but also in cows), ideally, longitudinal studies are preferred. Following each cow over time during different reproductive stages could give a better overview of the changes in the gut microbiota over time. Limitations and recommendations for future studies are not included as they should be.
Re:We have positive and negative controls during the DNA extraction and the library preparation. To minimize the difference between the samples within the group, we selected 6 cows for each group. Although this is somewhat limited, it is sufficient to obtain a preliminary result for further study. In the further studies, we will improve these limitations and recommendations.
Metadata and sequencing results should be made available in a public repository.
Re:According to the recent policy, we need to go through tedious procedures to upload the data.However, we can send the data to interested people via correspongding auther's email at any time.
Specific comments
Line 20: Why did you choose those acronyms to represent each group? What do the letters stand for?
Re:They are the contraction of the first word. However,we forgot how to spell the word primiparous,when we collected samples.So, we used Chinese pinyin abbreviations instead of primiparous. Here, BG=Big Breeding cattle, FT=First Pregnany,DC=Duoci,P=pregnancy,NP=no-pregnancy。
Line 56 -59: Please review the sentence, changes in the writing are required for clarity. Also, it would be advisable to include the results of previous studies made with cows.
Re:We modified the sentence and added the previous studies in dairy cows.
Line 86: Please provide more information about the study population, living conditions, farm practices, vaccination, environment, and diet. The age of each group should be specified as well as the body condition score and body weight. All these factors can influence gut microbiota composition. Also, for reproducibility, all these factors should be mentioned.
Re:All the cows are from the same farm, and have the same feeding processes and condition.The cows in one group were selected to ensure that they were at the same age, body condition, parity and similar body weight.We try to ensure that only pregnancy factor different in each group.
Line 97: How much faecal material was used for DNA extraction? Were they aliquoted before storage at -80, or were they thawed before DNA extraction? How much DNA was used for library preparation?
Re: All Fecal 24 fecal samples were obtained once from cow rectum content at the same day. Each sample were transferred to separate sterilized 2 mL tubes, and then stored immediately in liquid nitrogen. All samples were then transported to the laboratory and stored at -80 oC for further DNA extraction. The 2 mL of each faecal material was taken out from -80 oC refrigerator for DNA extraction at a time. About 100 ng of DNA was used for library preparation.
Line 112: Do you mean that the technical replicates were merged?
Re:Three replicates of PCR reaction were performed for each sample, and PCR products of the same sample were mixed for purification.
Line 123: did you perform rarefaction curves to assess if sequencing depth was appropriate?
Re: Yes, we do. And the curves flattens out for all samples.
Line 144: Spelling mistake in legend (umber versus number). Could you explain the statistics performed in figure 1, please? It was not specified in the methodology. Exact p-values should be specified.
Re:We modified the spelling mistake.Results of figure 1 was analyzed by one-way analysis of variance (one way ANOVA), followed by Bonferroni multiple comparisons test.The significance level is p-value <0.01.
Line 148: That number of genera and species probably doesn’t reflect the total number, as with 16S rRNA sequencing, not all genera and species can be identified.
Re:Yes,it is.We've analyzed the data that we could get from 16S rRNA sequencing.
Line 162: Two cladograms are shown for the FT group or there is a spelling mistake, and the third cladogram corresponds to DCNP.
Re:Yes, it is. We modified it.
Line 163: Visually inspecting figure 3, it is evident that DCNP_5 and _6 have a different profile compared to the other four cows, probably influencing the results, as only 6 cows were analysed by group. At least, at the phylum level. This is also probably reflected in figure 6, where it is evident that 4 cows cluster together whereas the other 2 clustered but separated from the first 4.
Re:Yes. It is the relative higher abundance of Euryarchaeota and the relative lower abundance of Fimicutes in these two samples.
Line 247: Pregnancy status. If Age and pregnancy are contributing factors, why do you think there were no differences between multiparous pregnant cows and nulliparous and primiparous cows, as they differ in age and pregnancy status? Scientific conclusions should not be based only on whether a p-value passes a specific threshold. The number of subjects per group is small. Based on figure 2, in general, alpha diversity was higher in those of a younger age. Could you explain how age influences diversity, please?
Re:The alpha diversity indexes of BG group are close to that of FT group, while the alpha diversity indexes of DCNP group are close to that of DCP. And based on the Jaccard and Bray-Curtis methods, the samples in the BG and FT groups tended to cluster together in accordance with PCoA results, While there is no significant difference between groups DCP vs. DCNP. While the differences of the alpha diversity indexes between BG group and FT group had the same trend with that between DCNP and DCP. For the above reasons, we speculated that age and pregnant are two important factors contributing to the species richness and diversity of fecal microbiota. The effecttion of age may be more due to whether calving has occurred.
Line 254: You mention that pregnancy is related to an increase in alpha diversity, could you explain the reason, please?
Re:The alpha diversity indexes of FT group higher than BG group, and DCP group higher than DCNP, except Coverage index. So we mention that pregnancy is related to an increase in alpha diversity. Which may be interpreted as increased nutrient requirements of the cow during lactation.
Line 267:You have used two different statistical methods for differential abundance. LEfSe was used for general comparison whereas pairwise comparisons were made using metagenomSeq (Based on figure 8). What was the reason? Currently, some people recommend the use of different statistical methods to find the most robust and consistent changes, but in this case, it is not clear why you decided to use two different statistical methods.
Re:LEfSe was usually used for general comparison,and pairwise comparisons were usually made using metagenomSeq. We used these two different common statistical methods to find more differential taxa or most robust and consistent taxa for further in-depth analysis.

Round 2
Reviewer 2 Report
Estimate editor,
Dear Editor,
I still feel the same way about this article, I remind you of my previous comments (below). The authors have modified paragraphs that other reviewers have proposed. I had proposed that they use another methodology to correlate microbiota with fertility. As they have not changed the methodology of the manuscript, I still say that for me it is not publishable.
I have reviewed the article “Comparison of fecal microbiota communities between primiparous and multiparous cows during nonpregnancy and pregnancy “ . After trying to make sense of the research, I feel compelled to reject it.
This study investigates the fecal microbiome composition between primiparous and multiparous cows during non-pregnancy and pregnancy to analyze the host-microbial balance at different stages.
They have detected that there are differences in the microbiota in pregnant cows compared to non-pregnant cows. And this conclusion is very poor, because the metabolic and hormonal status of a pregnant cow is different from the status of a non-pregnant cow. Therefore, it is normal for the microbiota to be different.
They should try to determine what consequences this change has at the level of intestinal microbiota on reproductive efficiency (in ovarian folliculogenesis, or at the level of embryonic development, etc...) Because such a change detected in the microbiota, it is possible that it is normal and necessary due to the pregnant status of the animal.
The title of the manuscript is very attractive, but when one studies the article, the title of the article does not correspond to the development of the article.
For all these reasons, I am obliged not to accept the manuscript, as it is presented. If the authors continue to develop this line of research, the implication of the microbiota in pregnant women on reproductive efficiency, a very interesting and publishable article will surely emerge.
Re:This study investigates the fecal microbiome composition between primiparous and multiparous cows during non-pregnancy and pregnancy to analyze the host-microbial balance at different stages.The results indicate that host-microbial interactions promote the adaptation to pregnancy.Of course,these findings are preliminary results.Further in-depth analysis is required on the basis of this study. And then, the results will benefit the development of probiotics or fecal transplantation for treating dysbiosis and preventing disease development during pregnancy.
Author Response
I still feel the same way about this article, I remind you of my previous comments (below). The authors have modified paragraphs that other reviewers have proposed. I had proposed that they use another methodology to correlate microbiota with fertility. As they have not changed the methodology of the manuscript, I still say that for me it is not publishable.
Re:The 24 sample sizes were not large enough for association analysis, but the difference between each group was statistically significant.
Reviewer 3 Report
Thank you so much for the revised version and for the responses to the comments. However, the manuscript still requires some adjustments to be suitable for publication.
It is essential that your responses to the comments are reflected in the manuscript. The selection criteria for the time points selected and the criteria of inclusion for the cows (e.g., uniform production records) should be clearly specified in the manuscript. A clear definition of inclusion and exclusion criteria helps define confounding factors and is vital for reproducibility. It is excellent that you selected cows from the same farm, with the same feeding processes, age, body condition, parity and similar body weight. Please specify it in the manuscript so that every reader knows that you controlled these potentially confounded factors.
Please specify in the manuscript that about 100 ng of DNA were used for library preparation and that you have used a one-way analysis of variance (one-way ANOVA), followed by the Bonferroni multiple comparisons test for figure 1.
You also mentioned in the response that you used negative and positive controls. However, in the manuscript, there is no description of the controls used and how and when they were used during the workflow. Controls are used to assess the level of contamination (negative), whereas positives are used to assess DNA extraction and sequencing efficiency. There is no analysis of the controls.
There is no description of the limitations of the study or recommendations for further studies. I agree with you this study could be considered a pilot study for future studies and a base for sample size calculations for the next studies. I also agree with you that if there is no function reported for a bacterium, it is difficult to assess the relevance of this finding. However, for example, you mentioned that Rikenellaceae_RC9_gut_group was associated with intestinal pathogens colonization and inflammation in mice gut. Why is this important during pregnancy? It is not only about mentioning the functions reported for a determined bacterium or group, but also to correlate these functions with the topic.
In line 408, you mentioned that the biggest microbiome trajectory changes occurred between nulliparous and primiparous cows, however, the PcoA plots show that they have the tendency to cluster together. Then in line 410, you mention that nulliparous cows have a higher diversity compared to primiparous cows but then you say that pregnancy increase diversity, so it is contradictory. According to the boxplots, it seems that alpha diversity is higher during pregnancy (nulliparous versus primiparous and between open multiparous cows versus pregnant multiparous cows). Also, the diversity is lower in multiparous cows (irrespective of their pregnancy status) compared to nulliparous and primiparous cows (lines 233-234). So, it seems that number of pregnancies can have an impact on alpha diversity.
In line 234, you mentioned that coverage was higher in the DCNP group, what is the meaning or relevance of this finding (lower diversity and higher coverage in the DCNP group)?
I agree with you that pregnancy and age are two important factors for diversity, and this is what you have found in your study. However, you need to provide an analysis (reasons) for these findings. Why does the diversity increase during pregnancy and decrease with age? Are the findings related to stability, longer or shorter exposure to environmental challenges, and maturity of the gut microbiome, or due to metabolic changes? For example, in your response to comments, you mention that effect of age is more related to calving and that the increase in alpha diversity during pregnancy, could be due to an increase in nutrient requirements during lactation. However, this analysis is not reported in the manuscript.
Please specify that pairwise comparisons were made using metagenomSeq for clarity.
What are your thoughts regarding the two outliers in the DCNP group (DCNP_5 and DCNP_6)?
In addition, bacterial names are italicized in the text, so they should be italicized in the figures as well (only figure 3 has the names in italics). In figure 2, you show a lot of percentages but some of them don’t have any name assigned at lower phylogenetic levels. In methods (section 2.3), all reagents should have the manufacturer. In line 143, the name of the kit is Biomics DNA microprep kit and the manufacturer is Zymo Research. The first Zymo research should be deleted. In some lines you write USA in others you write United States, please be consistent. Qubit is from Thermofisher for example.
Finally, the manuscript still has spelling and grammatical mistakes. For example, in line 98 (it should be maternal hormones), the sentences in lines 100 and 101 need to be corrected, line 109, line 145 (it should be were tested), line 147 (the variable region), line 181, line 208 (it should be number), lines 235 and 237 ( it should be Kruskal-Wallis test), line 312, line 407, line 418 (what is DDCNPNP?), among others.
Finally, different data repositories for raw data and metadata are available, such as the National Center for Biotechnology Information (NCBI), The European Bioinformatics Institute (EMBL-EBI), the DNA Data Bank of Japan (DDBJ) and the Genome Sequence Archive (GSA). I am not sure what you mean by difficult policies but if it is not possible to deposit the data in a public repository, at least, you should mention that data will be available upon request.
Author Response
Thank you so much for the revised version and for the responses to the comments. However, the manuscript still requires some adjustments to be suitable for publication.
It is essential that your responses to the comments are reflected in the manuscript. The selection criteria for the time points selected and the criteria of inclusion for the cows (e.g., uniform production records) should be clearly specified in the manuscript. A clear definition of inclusion and exclusion criteria helps define confounding factors and is vital for reproducibility. It is excellent that you selected cows from the same farm, with the same feeding processes, age, body condition, parity and similar body weight. Please specify it in the manuscript so that every reader knows that you controlled these potentially confounded factors.
Re:Thanks. We added these information in the manuscript.
Please specify in the manuscript that about 100 ng of DNA were used for library preparation and that you have used a one-way analysis of variance (one-way ANOVA), followed by the Bonferroni multiple comparisons test for figure 1.
Re:Thanks. We added these information in the manuscript.
You also mentioned in the response that you used negative and positive controls. However, in the manuscript, there is no description of the controls used and how and when they were used during the workflow. Controls are used to assess the level of contamination (negative), whereas positives are used to assess DNA extraction and sequencing efficiency. There is no analysis of the controls.
Re:We set up negative and positive controls from the beginning of DNA extraction. No contamination was found in the negative control, and the DNA extraction and sequencing efficiency of the positive control were high. Referring to the way materials and methods are written in most literature, we do not list this information.
There is no description of the limitations of the study or recommendations for further studies. I agree with you this study could be considered a pilot study for future studies and a base for sample size calculations for the next studies. I also agree with you that if there is no function reported for a bacterium, it is difficult to assess the relevance of this finding. However, for example, you mentioned that Rikenellaceae_RC9_gut_group was associated with intestinal pathogens colonization and inflammation in mice gut. Why is this important during pregnancy? It is not only about mentioning the functions reported for a determined bacterium or group, but also to correlate these functions with the topic.
Re:Yes. We plan to further screen for differences in gut microbes between early parities and late parities with reproductive disorders in dairy cow.
In line 408, you mentioned that the biggest microbiome trajectory changes occurred between nulliparous and primiparous cows, however, the PcoA plots show that they have the tendency to cluster together. Then in line 410, you mention that nulliparous cows have a higher diversity compared to primiparous cows but then you say that pregnancy increase diversity, so it is contradictory. According to the boxplots, it seems that alpha diversity is higher during pregnancy (nulliparous versus primiparous and between open multiparous cows versus pregnant multiparous cows). Also, the diversity is lower in multiparous cows (irrespective of their pregnancy status) compared to nulliparous and primiparous cows (lines 233-234). So, it seems that number of pregnancies can have an impact on alpha diversity.
Re:The PcoA plots show that BG and FT have the tendency to cluster together, and distinct with the other two groups.Neither BG or FT group had ever given birth. So, the biggest microbiome trajectory changes occurred between nulliparous and primiparous cows. According to the boxplots, the diversity is lower in multiparous cows(BG vs.DCNP, FT vs.DCP), and the alpha diversity is higher during pregnancy (FT vs.BG, DCP vs.DCNP).
In line 234, you mentioned that coverage was higher in the DCNP group, what is the meaning or relevance of this finding (lower diversity and higher coverage in the DCNP group)?
Re:Coverage means the Good's overage. The Good's coverage estimate was calculated to assess the "percentage diversity" captured by sequencing. In our study, the Good's coverage for all samples were >0.95.
I agree with you that pregnancy and age are two important factors for diversity, and this is what you have found in your study. However, you need to provide an analysis (reasons) for these findings. Why does the diversity increase during pregnancy and decrease with age? Are the findings related to stability, longer or shorter exposure to environmental challenges, and maturity of the gut microbiome, or due to metabolic changes? For example, in your response to comments, you mention that effect of age is more related to calving and that the increase in alpha diversity during pregnancy, could be due to an increase in nutrient requirements during lactation. However, this analysis is not reported in the manuscript.
Re:We've added this analysis.
Please specify that pairwise comparisons were made using metagenomSeq for clarity.
Re:We've modified.
What are your thoughts regarding the two outliers in the DCNP group (DCNP_5 and DCNP_6)?
Re:The variation of fecal microorganism may be greater due to the difference of body recovery state of cows in the empty period.
In addition, bacterial names are italicized in the text, so they should be italicized in the figures as well (only figure 3 has the names in italics). In figure 2, you show a lot of percentages but some of them don’t have any name assigned at lower phylogenetic levels. In methods (section 2.3), all reagents should have the manufacturer. In line 143, the name of the kit is Biomics DNA microprep kit and the manufacturer is Zymo Research. The first Zymo research should be deleted. In some lines you write USA in others you write United States, please be consistent. Qubit is from Thermofisher for example.
Re:The pictures are drawn by specialized data analysts in accordance with our research ideas at one time, which is difficult to adjust.Figure 2 is too crowded to write the names.We have modified the others.
Finally, the manuscript still has spelling and grammatical mistakes. For example, in line 98 (it should be maternal hormones), the sentences in lines 100 and 101 need to be corrected, line 109, line 145 (it should be were tested), line 147 (the variable region), line 181, line 208 (it should be number), lines 235 and 237 ( it should be Kruskal-Wallis test), line 312, line 407, line 418 (what is DDCNPNP?), among others.
Re:These errors may be caused by the revision state in the Microsoft word.We have checked and modified one by one.
Finally, different data repositories for raw data and metadata are available, such as the National Center for Biotechnology Information (NCBI), The European Bioinformatics Institute (EMBL-EBI), the DNA Data Bank of Japan (DDBJ) and the Genome Sequence Archive (GSA). I am not sure what you mean by difficult policies but if it is not possible to deposit the data in a public repository, at least, you should mention that data will be available upon request.
Re:Uploading data to a public database requires a lot of leadership approval. We chose the option that data will be available upon request, when we submitted.

Round 3
Reviewer 2 Report
No coments
Author Response
We carefully improved and checked the whole manuscript.
Reviewer 3 Report
Thank you so much for the revised version and for the responses to the comments. I would recommend some final adjustments to be suitable for publication.
The use of negative and positive controls is highly recommended and if they were used, they must be described in the methodology section. The fact that many studies have not used them and/or reported them is not a reason to not report them in the manuscript. In fact, currently, more papers are reporting them as this is considered part of best practices in the gut microbiome. A brief description of the controls used with the results obtained should be added to the methodology.
I would also recommend that you specify the age, BCS and weight of the cows, if possible.
In line 159 you said that the relative abundance of the phylum Euryarchaeota was 15.42% in all metagenomic libraries, whereas in line 248, you said that the abundance was 0.25%. Which one is the real value?
In line 191, you said that you used ANOSIM, did you compare the four groups at the same time and then you performed pairwise comparisons? You said that BG and FT tended to cluster together, but only the comparison between DCP vs. DCNP was not significantly different.
There are still spelling and grammatical mistakes in the manuscript, i.e., line 61, dairy cows, line 57: was significant differences (this is not correct), line 257, line 268, among others.
Again, the most important part of the manuscript is the analysis, and you should emphasise this aspect. In lines 262-264, you provide important information but don’t specify how calving and how an increase in nutrient requirements lead to increase diversity. Diversity indexes were higher in younger cows compared to older cows, irrespective of their pregnancy status, so other reasons besides calving should be considered.
In your previous response, you said: “The PcoA plots show that BG and FT have the tendency to cluster together, and distinct with the other two groups. Neither BG or FT group had ever given birth. So, the biggest microbiome trajectory changes occurred between nulliparous and primiparous cows”. This isn't very clear. The primiparous cows in this study are the ones in the FT group and the nulliparous are in the BG group. The other groups have more than three lactation periods.
In line 276, you said that the most representative taxa were associated with energy metabolism and inflammation. Were these taxa more abundant in younger or older cows or in pregnant versus non-Pregnant? What is the significance of these findings at the physiological level?
Author Response
1, The use of negative and positive controls is highly recommended and if they were used, they must be described in the methodology section. The fact that many studies have not used them and/or reported them is not a reason to not report them in the manuscript. In fact, currently, more papers are reporting them as this is considered part of best practices in the gut microbiome. A brief description of the controls used with the results obtained should be added to the methodology.
Re:We added relevant information:negative control (DNA free water) and positive control (16S Universal E29).
2, I would also recommend that you specify the age, BCS and weight of the cows, if possible.
Re:We evaluated body condition and measured body size. And then we selected individuals within each group with similar values.
3, In line 159 you said that the relative abundance of the phylum Euryarchaeota was 15.42% in all metagenomic libraries, whereas in line 248, you said that the abundance was 0.25%. Which one is the real value?
Re:In our study the relative abundance of the phylum Euryarchaeota was 15.42%, in others it was 0.25%.
4, In line 191, you said that you used ANOSIM, did you compare the four groups at the same time and then you performed pairwise comparisons? You said that BG and FT tended to cluster together, but only the comparison between DCP vs. DCNP was not significantly different.
Re:Combined with PCoA-pictures and statistical tests, this is indeed the case.
5, There are still spelling and grammatical mistakes in the manuscript, i.e., line 61, dairy cows, line 57: was significant differences (this is not correct), line 257, line 268, among others.
Re:We carefully checked the whole manuscript.
6, Again, the most important part of the manuscript is the analysis, and you should emphasise this aspect. In lines 262-264, you provide important information but don’t specify how calving and how an increase in nutrient requirements lead to increase diversity. Diversity indexes were higher in younger cows compared to older cows, irrespective of their pregnancy status, so other reasons besides calving should be considered.
Re:The first birth is the most important physiological change in a cow's life and pregnancy increases metabolism.
7, In your previous response, you said: “The PcoA plots show that BG and FT have the tendency to cluster together, and distinct with the other two groups. Neither BG or FT group had ever given birth. So, the biggest microbiome trajectory changes occurred between nulliparous and primiparous cows”. This isn't very clear. The primiparous cows in this study are the ones in the FT group and the nulliparous are in the BG group. The other groups have more than three lactation periods.
Re:The FT group was at the 4th month of first pregnancy.They haven't experienced their first calving yet.
8, In line 276, you said that the most representative taxa were associated with energy metabolism and inflammation. Were these taxa more abundant in younger or older cows or in pregnant versus non-Pregnant? What is the significance of these findings at the physiological level?
Re:The taxa associated with energy metabolism were aboundant in pregnant vs non-pregnant cows. And the taxa associated with inflammation were aboundant in older vs younger cows. Pregnancy enhances metabolism, and more than 3 parities may cause inflammation of reproductive tract.